# Synthesis of Yttria-Stabilized Zirconia Nanospheres from Zirconium-Based Metal–Organic Frameworks and the Dielectric Properties

**DOI:** 10.3390/nano13010028

**Published:** 2022-12-21

**Authors:** Hyun Woo Park, Eunyeong Cho, Yun Zou, Sea Hoon Lee, Jae Ryung Choi, Sang-Bok Lee, Kyeongwoon Chung, Se Hun Kwon, Jeonghun Kim, Hee Jung Lee

**Affiliations:** 1Composites Research Division, Korea Institute of Materials Science (KIMS), 797 Changwon-daero, Sungsan-Gu, Changwon, Gyeongsangnam-do 51508, Republic of Korea; 2School of Materials Science and Engineering, Pusan National University, 2, Busandaehak-ro 63 beon-gil, Geumjeong-gu, Busan 46241, Republic of Korea; 3Department of Chemical and Biomolecular Engineering, Yonsei University, 50 Yonsei-ro, Seodaemun-gu, Seoul 03722, Republic of Korea; 4Department of Space and Defense Materials, Korea Institute of Materials Science (KIMS), 797 Changwon-daero, Sungsan-gu, Changwon-si, Gyeongsangnam-do 51508, Republic of Korea; 5Department of Advanced Materials Engineering, University of Science and Technology (UST), 217 Gajeong-ro, Yuseong-gu, Daejeon 34113, Republic of Korea; 6Department of Biofibers and Biomaterials Science, Kyungpook National University, 80, Daehak-ro, Buk-gu, Daegu 41566, Republic of Korea

**Keywords:** metal–organic framework, yttria-stabilized zirconia, ceramic, 5G communication, complex permittivity

## Abstract

Yttria-stabilized zirconia (YSZ) nanospheres were synthesized by calcination at 900 °C after the adsorption of Y^3+^ ions into the pores of a zirconium-based metal–organic framework (MOF). The synthesized 3YSZ (zirconia doped with 3 mol% Y_2_O_3_), 8YSZ (8 mol% Y_2_O_3_), and 30YSZ (30 mol% Y_2_O_3_) nanospheres were found to exhibit uniform sizes and shapes. Complex permittivity and complex permeability were carried out in K-band (i.e., 18–26.5 GHz) to determine their suitability for use as low-k materials in 5G communications. The real and imaginary parts of the permittivity of the sintered 3YSZ were determined to be 21.24 and 0.12, respectively, while those of 8YSZ were 22.80 and 0.16, and those of 30YSZ were 7.16 and 0.38. Control of the real part of the permittivity in the sintered YSZ was facilitated by modifying the Y_2_O_3_ content, thereby rendering this material an electronic ceramic with potential for use in high-frequency 5G communications due to its excellent mechanical properties, high chemical resistance, and good thermal stability. In particular, it could be employed as an exterior material for electronic communication products requiring the minimization of information loss.

## 1. Introduction

As the demand for 5G communications increases for applications such as big data, artificial intelligence, autonomous driving, the Internet-of-Things (IoT), and telemedicine, the frequency used for processing large amounts of information gradually increases accordingly, thereby highlighting the importance of developing superior dielectric materials [1,2,3,4]. In this context, the millimeter-wave (MM-Wave) band at ~30 GHz possesses a short wavelength and can transmit large-scale information at high speed through a wide bandwidth. Therefore, as the propagation loss increases toward the high-frequency band, the dielectric material for 5G communication must have low real and imaginary parts of permittivity to minimize the propagation loss. For this reason, cables, antennas, and exterior materials for devices for 5G communications must also be newly developed. In particular, yttria-stabilized zirconia (YSZ), which is widely used as an exterior material for electronic devices based on its advantages such as high mechanical strength and thermal and chemical stability, needs a great deal of improvement because of its high real part of the permittivity in the high-frequency band and high propagation loss.

YSZ powder is generally synthesized through the solid phase, liquid phase, and gas phase methods, among which the liquid phase method is mainly used in industry due to its cost-effectiveness and ease of processing [5,6,7]. This method is a process in which a precursor is synthesized through a precipitation or gelation reaction of a raw material such as a metal chloride or nitride, and a crystalline ceramic powder is obtained through drying and calcination of the precursor. However, the crystal growth and aggregation induced by this process prevent particle size and shape control [7]. To solve this problem, a grinding process such as ball milling may be added, but it is difficult to control the phase due to phase transformation caused by stress. The crystalline phase that directly affects the properties of YSZ depends on the content of Y_2_O_3_. In general, tetragonal YSZ is synthesized by doping ZrO_2_ with 3 mol% of Y_2_O_3_ and is mainly used as a structural or refractory material due to its high Vickers hardness, fracture toughness, and heat resistance. On the other hand, cubic YSZ is synthesized by doping ZrO_2_ with 8 mol% or more of Y_2_O_3_, and it has high ionic conductivity and low thermal conductivity, so it is suitable for application as an electrolyte for SOFC or insulators. Therefore, it is necessary to develop an efficient YSZ powder manufacturing process that precisely controls the particle size, shape, crystalline phase, and composition so that it can be customized according to the application purpose.

Metal–organic frameworks (MOFs) are coordination polymer compounds with a periodic and nanoporous structure, formed from the coordination of organic ligands to metal ions or clusters. The numerous functional organic ligands and metal ions, as well as the large specific surface area of MOFs, facilitate the chemical and physical modifications that allow these functional nanoporous materials to be tailored toward diverse applications, including catalysis, sensors, and gas storage [8,9,10,11,12,13,14]. Moreover, the particle sizes, shapes, and compositions of MOFs can be readily modified by controlling the crystal growth mechanism during MOF synthesis. It has been reported that this offers great potential for use as a templated synthetic technique to prepare metal oxides [15,16].

We herein report a simple method for the synthesis of YSZ nanospheres with a uniform size and shape, along with a controlled phase composition, using MOF-801 as the template (see Figure 1). MOF-801, a zirconium-based MOF, is composed of octahedral Zr_6_O_4_(OH)_4_ secondary building units (SBUs) and fumaric acid ligands. Each Zr_6_ metal cluster coordinates to 12 fumarate ions to form a regular three-dimensional nanoporous structure that is extremely stable, owing to the high coordination number and strong covalent bonds between the zirconium ions and fumarate [17,18]. YSZ precursors prepared by the adsorption of Y^3+^ ions (i.e., 3, 8, and 30 mol%) into the periodic nanoporous structure of MOF-801 are converted into YSZ nanospheres through calcination at 900 °C in air. The phase composition of YSZ nanospheres is precisely controlled by varying the yttrium content of the MOF-801, while a uniform distribution of Y^3+^ ions in the YSZ precursor is ensured by subjecting the mixture of MOF-801 and the yttrium source to a gentle ball milling process. In addition, the Y^3+^ ions inside the pores of MOF-801 inhibit crystal growth during calcination, such that the particle size of MOF-801 shrinks uniformly while maintaining the particle shape. Finally, the YSZ nanospheres are also sintered under air at 1450 °C to study their complex permittivity and permeability properties, as well as to evaluate the performance of the electronic ceramics materials through measurement of their relative densities, hardness values, and thermal conductivity.

## 2. Materials and Methods

### 2.1. Materials

MOF-801 was synthesized using ZrOCl_2_·8H_2_O (98.0%, Sigma-Aldrich, Saint Louis, MO, USA), fumaric acid (≥99.0%, Sigma-Aldrich), formic acid (≥95.0%, Sigma-Aldrich), and N, N-dimethylformamide (DMF, 99.5%, Samchun Chemical, Seoul, Republic of Korea), and washed with ethanol (99.0%, Samchun Chemical). The YSZ precursor was synthesized using DI water and Y(NO_3_)_3_·6H_2_O (99.8%, Sigma-Aldrich) as a Y^3+^ ion source. A green compact was prepared by mixing poly(vinyl alcohol) (PVA) (Samchun Chemical) with YSZ nanospheres as a binder. All chemicals were purchased from commercial suppliers and used as received, without further purification.

### 2.2. Synthesis of MOF-801 and YSZ Precursors

ZrOCl_2_·8H_2_O (40.0 g, 124 mmol) and fumaric acid (16.5 g, 142 mmol) were added to DMF (700 mL) prior to the addition of formic acid (145 mL) and heated at 120 °C under stirring (230 rpm) for 16 h. The obtained white precipitate was centrifuged, washed 3 times with ethanol, and dried in an oven at 100 °C for 3 h to obtain MOF-801 with a particle size of 100 nm. Then, the synthesized MOF-801 (31 g) and Y(NO_3_)_3_·6H_2_O (3.2409 g, 9.1121 g, and 44.9096 g for 3YSZ, 8YSZ, and 30YSZ, respectively) were mixed in DI water (200 mL) under ultrasonication for 10 min. The solvent was then evaporated at 95 °C and the obtained product was dried in an oven at 100 °C for 3 h. The amounts of MOF-801 and Y(NO_3_)_3_·6H_2_O were determined by theoretical calculations of the ZrO_2_ and Y_2_O_3_ content of the YSZ nanospheres.

### 2.3. Synthesis of YSZ Nanospheres

The 3YSZ precursor (31 g), 8YSZ precursor (31 g), and 30YSZ precursor (31 g) were calcined separately at 900 °C for 2 h in air, with a heating rate of 5 °C/min, to obtain 3YSZ, 8YSZ, and 30YSZ nanospheres, respectively.

### 2.4. Preparation of Sintered YSZ

The 3YSZ, 8YSZ, and 30YSZ nanospheres (12.6 g) were separately mixed with an aqueous solution of PVA (1.4 g, 2 wt.%) using a mortar, and the mixture was uniaxially die-pressed (20 MPa, 2 min) prior to cold isostatic pressing (200 MPa, 5 min). The obtained green compact was heated to 1450 °C at a heating rate of 3 °C/min, and then sintered by holding this temperature for 3 h.

### 2.5. Characterization

The phase compositions were evaluated by X-ray diffraction (XRD) spectroscopy (D/max2500, Rigaku, Tokyo, Japan) with a Cu K_α_ (λ = 0.15406 nm) radiation source. The XRD patterns were recorded at a scanning speed of 2°/min. The morphologies were observed by field emission scanning electron microscopy (FE-SEM, JSM-6700F, JEOL, Tokyo, Japan) and transmission electron microscopy (TEM, JEM-2100F, JEOL). The Brunauer–Emmett–Teller (BET) specific surface area using nitrogen gas was measured using volumetric adsorption equipment (BELSORP MAX, MicrotracBEL, Osaka, Japan). MOF-801 powder was pretreated at 150 °C under vacuum for 16 h, and YSZ powder for 3 h, to remove absorbed gases and solvents prior to measuring the specific surface area. The yield (wt.%) of ZrO_2_ derived from MOF-801 was determined using thermogravimetric analysis (TGA, TGA55, TA instruments, New Castle, DE, USA). The composition of YSZ was determined by wavelength-dispersive X-ray fluorescence (WDXRF, S8 TIGER, BRUKER, Billerica, MA, USA). The surface element analysis was performed on an X-ray photoelectron spectroscopy instrument (XPS, Thermo Fisher Scientific, Waltham, MA, USA) using monochromatic Al K_α_ radiation (spot size = 400 μm). The thermal conductivity was analyzed using a thermal conductivity analyzer (LFA 467 HyperFlash, NETZSCH, Hanau, Germany). The Vickers hardness values were determined using a Micro Vickers hardness tester (HM-220C, Mitutoyo, Tokyo, Japan). In the Vickers hardness test, a diamond pyramid indenter with an angle between opposite faces of 136° was pressed into the specimen with a test force of 0.1 kgf.

## 3. Results and Discussion

### 3.1. Synthesis of MOF-801

Figure 1a shows that the average particle size of the synthesized spheroidal MOF-801 used to prepare YSZ nanospheres was approximately 100 nm. The XRD analysis of the prepared MOF-801 confirmed the presence of the MOF-801 crystal structure, which was in good agreement with the simulated pattern calculated based on a previously reported MOF-801 single crystal [18] (Figure 1b). In addition, as shown in Figure 1c, the nitrogen adsorption/desorption isotherm of MOF-801 showed typical Type I behavior, while the BET specific surface area and the total pore volume of the sample were determined to be 1078 m^2^ g^−1^ and 0.51 cm^3^ g^−1^, respectively, demonstrating the nanoporous nature of MOF-801. The TGA curve presented in Figure 1d reveals that MOF-801 undergoes a four-stage decomposition process in air: (1) 0.8% weight loss below 100 °C, corresponding to dehydration of the MOF-801; (2) 3.9% weight loss between 100 and 260 °C, associated with the desolvation process; (3) 31.2% weight loss between 260 and 450 °C, reflecting the decomposition of the fumarate ligand; (4) 8.4% weight loss between 450 and 900 °C, owing to the loss of carbon dioxide. No additional weight loss was observed above 900 °C, indicating that the complete conversion of MOF-801 into ZrO_2_ occurs below this temperature, with a yield of ~55.7 wt.%.

### 3.2. YSZ Nanospheres Derived from MOF-801-Based Precursor

The synthesized YSZ precursor retained both the size and shape of the initial MOF-801 (Appendix A), indicating that MOF-801 acts as a suitable template for controlling the formation of YSZ nanospheres with defined nanostructures and uniform morphologies. Although it was found that the introduction of Y^3+^ ions to MOF-801 did not affect the original crystalline structure (Appendix A), the BET specific surface area and the total pore volume decreased as the level of Y^3+^ ion doping increased (Appendix A). More specifically, the specific surface areas and total pore volumes were determined to be as follows: 767 m^2^ g^−1^ and 0.49 cm^3^ g^−1^ for the 3YSZ precursor; 515 m^2^ g^−1^ and 0.38 cm^3^ g^−1^ for the 8YSZ precursor; and 17.3 m^2^ g^−1^ and 0.04 cm^3^ g^−1^ for the 30YSZ precursor. These results confirm that Y^3+^ ion incorporation at 30 wt.% can be achieved by adsorption into the MOF-801 pores. Thus, 3YSZ, 8YSZ, and 30YSZ nanospheres were synthesized from the YSZ precursor following the adsorption of Y^3+^ ions into the pores of MOF-801 and subsequent heat treatment at 900 °C, as described above. The synthesized YSZ nanospheres were found to be highly homogeneous and spherical in shape, with an average particle diameter of ~60 nm; these nanospheres maintained their shapes and slightly shrank at the particle size of MOF-801 with a diameter of ~100nm (Figure 2). Figure 2g–i show the distributions of Y, Zr, and O in the YSZ nanospheres, and it is evident that the Y^3+^ and Zr_4+_ ions are homogeneously distributed throughout the YSZ nanospheres. In addition, the number of Y^3+^ ions clearly increases upon increasing the Y_2_O_3_ content.

The XRD patterns of the synthesized YSZ nanospheres (Figure 3) reveal that the 3YSZ nanospheres consist of a pure tetragonal phase (JCPDS No. 48-0224), while the 8YSZ and 30YSZ nanospheres consist of the fully cubic phase (JCPDS No. 30-1468) [19,20]. Moreover, the nitrogen adsorption/desorption isotherms of all prepared YSZ nanospheres exhibit typical Type II behavior, indicating the non-porous nature of the YSZ nanospheres (Appendix A), which is consistent with their specific surface areas and total pore volumes (18 m^2^ g^−1^/0.22 cm^3^ g^−1^, 19 m^2^ g^−1^/0.22 cm^3^ g^−1^, and 7 m^2^ g^−1^/0.07 cm^3^ g^−1^ for 3YSZ, 8YSZ, and 30YSZ, respectively).

The Y_2_O_3_ content of the prepared YSZ nanospheres was evaluated by WFXRF, as presented in Table 1, wherein it can be seen that the obtained results are relatively consistent with the intended doping levels. Moreover, the EDX spectra show an obvious increase in the intensity of the Y^3+^ ion peak at ~14.9 keV upon increasing the Y_2_O_3_ doping amount (Appendix A). This result is in good agreement with the STEM-EDX mapping images presented in Figure 2g–i.

Figure 4 displays the XPS spectra of the synthesized YSZ nanospheres. The Zr 3d XPS spectra exhibit two peaks at 182.0 and 184.0 eV, corresponding to the 3d_5/2_ and 3d_3/2_ of the Zr^4+^ ions, respectively (Figure 4a,d,g). The Y 3d XPS spectra confirm the presence of Y^3+^ ions, showing two peaks at 157.3 and 159.2 eV, representing Y 3d_5/2_ and Y 3d_3/2_, respectively (Figure 4b,e,h). In addition, characteristic peaks corresponding to the lattice oxygen (metal–oxygen bond), surface oxygen vacancies, and adsorbed O_2_ were observed at 530.0, 531.4, and 532.3 eV, respectively, in the O 1s spectra (Figure 4c,f,i) [21]. Furthermore, the atomic compositions of the three types of nanospheres were determined to be as follows: Zr 3d = 56.54%, Y 3d = 5.32%, and O 1s = 38.14% for 3YSZ; Zr 3d = 50.65%, Y 3d = 11.54%, and O 1s = 37.81% for 8YSZ; and Zr 3d = 30.22%, Y 3d = 32.14%, and O 1s = 37.64% for 30YSZ. These correspond with the WDXRF and EDX spectra results (Table 1 and Appendix A).

The crystal structure and grain growth behaviors of the YSZ nanospheres were investigated by comparing their high-resolution transmission electron microscopy (HRTEM) images and selected area electron diffraction (SAED) patterns with their XRD patterns (Figure 3) [22,23,24]. As a result, the lattice spacings and lattice planes of the YSZ nanospheres were 0.30 nm (Figure 5d–f), which corresponds to the (101) lattice plane of the tetragonal phase in the 3YSZ nanospheres (Figure 5a) and the (111) lattice plane of the cubic phase in the 8YSZ and 30YSZ nanospheres (Figure 5b,c). In addition, the lattice planes identified from the SAED patterns (Figure 5g–i) are consistent with the XRD patterns shown in Figure 3. The (101), (110), (112), (200), (103), and (211) lattice planes in the 3YSZ nanospheres (Figure 5g) are associated with the tetragonal YSZ lattice planes, while the (111), (220), (311), (200), (103), and (211) lattice planes in the 8YSZ and 30YSZ nanospheres (Figure 5h,i) are correlated with the cubic YSZ lattice planes. As shown in Figure 5g–i, the SAED pattern changes as the Y_2_O_3_ content of the YSZ nanospheres is increased. More specifically, a ring shape is adopted, which suggests that increasing the Y_2_O_3_ content induce the formation of fine polycrystal grains. This change was verified by TEM observations of the polycrystalline YSZ nanospheres (Figure 5a–c). The crystal sizes of the 3YSZ, 8YSZ, and 30YSZ nanospheres were measured as 16.70, 13.01, and 10.01 nm, respectively. The reduced crystal sizes of the 3YSZ, 8YSZ, and 30YSZ nanospheres were attributed to the increasing Y_2_O_3_ content.

### 3.3. Properties of Sintered YSZ

The synthesized 3YSZ, 8YSZ, and 30YSZ nanospheres were prepared as green bodies, and sintered bodies were prepared by heat treatment at 1450 °C. Importantly, the YSZ sintered bodies prepared herein were obtained as zirconia nanospheres containing uniformly distributed Y_2_O_3_ particles, without the requirement of an additional sintering agent. The XRD patterns of the sintered YSZ nanospheres presented in Figure 6 show that the sintered body of the synthesized 3YSZ nanospheres is primarily composed of a tetragonal phase, although a small monoclinic phase is also observed, demonstrating that a phase transformation occurred during the sintering process, in which a small amount of the tetragonal phase transformed into a monoclinic phase. Conversely, the sintered bodies of 8YSZ and 30YSZ nanospheres retained their cubic phase after sintering. It was also considered that, in addition to the above-described phase transformation, grain growth and coarsening also occur during the sintering process.

Indeed, the SEM images presented in Appendix A show that the grain sizes increased significantly for all specimens; the sintered 8YSZ and 30YSZ bodies exhibited the largest and smallest grains sizes, respectively. These results indicate that the sintering of the 8YSZ nanospheres resulted in extensive grain growth, which benefits the densification of the green body by removing pores, because a larger grain size results in fewer pores inside the body. Conversely, grain growth in the 30YSZ nanospheres was far less extensive, owing to the strong inhibition of the ZrO_2_ grain growth by the high Y_2_O_3_ content, ultimately leading to a more porous sintered body in 30YSZ. Moreover, the morphologies of the sintered YSZ bodies were consistent with the relative densities of the three samples, as can be seen in Figure 7.

The relative density, Vickers hardness, and thermal conductivity of the sintered YSZ bodies were evaluated, as shown in Figure 7. More specifically, the 3YSZ and 8YSZ sintered bodies possessed dense structures with a relative density of ~99.5%, while the 30YSZ sintered body exhibited a porous structure with a relative density of 52.5%. The Vickers hardness values of the 3YSZ, 8YSZ, and 30YSZ sintered bodies were 8.0, 10.8, and 0.7 GPa, respectively. These results correlate with the relative densities, wherein the porous 30YSZ sintered body with the lowest relative density of 52.5% possesses the lowest hardness. Furthermore, the thermal conductivity of the 3YSZ, 8YSZ, and 30YSZ sintered bodies at 25 °C was 1.83, 2.63, and 0.77 W/mK, respectively, and it was found that the thermal conductivity was maintained even at 200 °C (Appendix A).

Finally, Figure 8 indicates that the real parts of the permittivity were determined to be 21.24 and 22.80 for the 3YSZ and 8YSZ sintered bodies, respectively, at the center frequency of the K-band (i.e., 22.25 GHz, 18.0–26.5 GHz). This result indicates that the real part of the permittivity increased as the concentration of Y_2_O_3_ increased. However, in the case of the 30YSZ sintered body, the real part of the permittivity decreased dramatically to 7.16 due to the large volume of air voids, with a low real part of the permittivity of 1.0. It is largely developed within the sintered body caused by the abovementioned characteristic of 30YSZ, leading to poor processability. Moreover, the real and imaginary parts of the permeability of the 3YSZ, 8YSZ, and 30YSZ sintered bodies were 1.0 and 0.0 in the K-band frequency, which demonstrates that the YSZ sintered bodies are non-magnetic materials at this frequency. As a result, the real part of the permittivity can be controlled by manipulating the Y_2_O_3_ content of the YSZ nanospheres.

## 4. Conclusions

A novel and simple method for the synthesis of yttria-stabilized zirconia (YSZ) nanospheres was developed, which employs a zirconium-based metal–organic framework (MOF, i.e., MOF-801, Zr_6_O_4_(OH)_4_(fumarate)_6_) as a template to effectively control the particle size, shape, composition, and phase of the product for tailored application purposes. The spheroidal MOF-801 samples (100 nm particle diameter) were doped with various levels of Y^3+^ content prior to their conversion into YSZ nanospheres by means of calcination. As a result, it was found that the particles shrunk uniformly to an average particle size of 60 nm while retaining their spherical shape. During the preparation of the YSZ precursors, the periodic nanoporous structure of MOF-801 allowed a precise amount of Y^3+^ ions to be homogeneously distributed throughout the particles. Thus, three different YSZ nanospheres were prepared by doping different amounts of Y_2_O_3_ (i.e., 3YSZ = 3 mol% Y_2_O_3_, 8YSZ = 8 mol% Y_2_O_3_, and 30YSZ = 30 mol% Y_2_O_3_), and these YSZ nanospheres were found to exhibit tetragonal (3YSZ) and cubic phases (8YSZ and 30YSZ). This synthetic method is expected to be applicable to a wide range of YSZ materials for use in various applications, owing to the ability to easily adjust the particle size and yttrium loading into the MOF-801 template. In particular, YSZ nanospheres and YSZ sintered bodies manufactured using MOF-801 as a template have great potential for the synthesis of low-k materials for large-scale information processing in 5G communication applications, since the permittivity characteristics of these materials can be easily controlled by varying the content of Y_2_O_3_.

## Data Availability

Not applicable.

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
