# Peer review of "Synthesis of Yttria-Stabilized Zirconia Nanospheres from Zirconium-Based Metal–Organic Frameworks and the Dielectric Properties"

_nanomaterials, 2022, doi:10.3390/nano13010028_

Round 1
Reviewer 1 Report
There are multiple references in the manuscript to figures S1-S8, which however are not included. These are often confused with the actual captions of Figures 1 to 5 that are included in the manuscript. This makes it nearly impossible to follow the arguments of the authors, since the supportive evidence is either not provided or it is not clear which figure they refer to.
The whole manuscript needs to be completely re-written. All obsolete references to figures that are not included in the paper need to be removed. Also there are issues with the style. The authors have to make sure that the presentation is concise and logical.
The value of 1.02 for the real part of the relative permittivity of 30YSZ in the Abstract is incorrect, please correct.
Please use real and imaginary parts of the relative permittivity instead of "dielectric constant".
Reviewer 2 Report
Dear authors,
The following changes need to be made:
1. In the introduction, you must add alternative methods for synthesizing similar compounds and indicate the advantages and disadvantages of the way you use compared to other methods.
2. Paragraph 2 It is necessary to add the manufacturer and the country of production of the materials and equipment used.
3. Section 2.5 What scanning speed on the XRD was used?
4. BET method: What gas was used?
5. XPS: Specify monochromator type, and beam diameter. Have the samples been pre-coated?
6. Microhardness measurement: Specify the type of indenter and the load used.
Round 2
Reviewer 1 Report
The manuscript is significantly improved, however the authors seem to be unaware of the meaning of the terms:
"complex permittivity" is a complex function that has real and imaginary part
"dielectric constant" is sometimes to denote the real part of the complex permittivity. Since, it is not really a constant but a function of the frequency, "dielectric constant" should not be used. Instead, it should be referred to as the "real part of the (complex) permittivity".
"dielectric loss" is often used to denote the imaginary part of the complex permittivity.
Please use the correct terms in the manuscript. If the real part of the permittivity is referred to, then "real part of the permittivity" should be used. If the imaginary part of the permittivity is referred to, then "dielectric loss" or "the imaginary part of the permittivity should be used". Please correct all instances of incorrect use (line 29, 45, 51, 292, 295, 299).
Line 298 should read: "...decreased dramatically by to 7.16 and 0.38, respectively..." However, the dielectric loss seems to have increased from 0.12 to 0.38. Therefore, the sentence should be edited to reflect the actual change in the numbers.
